# A Diameter Measurement Method of Red Jujubes Trunk Based on Improved PSPNet

Yichen Qiao [1], Yaohua Hu [2,*], Zhouzhou Zheng [1], Zhanghao Qu [1], Chao Wang [1], Taifeng Guo [1] and Juncai Hou [1,*]

1. College of Mechanical and Electronic Engineering, Northwest A&F University, Yangling 712100, China; qiaoyichen@nwafu.edu.cn (Y.Q.); zhengzz@nwafu.edu.cn (Z.Z.); qzhh@nwafu.edu.cn (Z.Q.); w2021055899@nwafu.edu.cn (C.W.); gtf2021@nwafu.edu.cn (T.G.)
2. College of Optical, Mechanical and Electrical Engineering, Zhejiang A&F University, Hangzhou 311300, China
* Correspondence: huyaohua@zafu.edu.cn (Y.H.); houjuncai@nwsuaf.edu.cn (J.H.); Tel.: +86-15291680166 (Y.H.); +86-18792954818 (J.H.)

**Abstract:** A trunk segmentation and a diameter measurement of red jujubes are important steps in harvesting red jujubes using vibration harvesting robots as the results directly affect the effectiveness of the harvesting. A trunk segmentation algorithm of red jujubes, based on improved Pyramid Scene Parsing Network (PSPNet), and a diameter measurement algorithm to realize the segmentation and diameter measurement of the trunk are proposed in this research. To this end, MobilenetV2 was selected as the backbone of PSPNet so that it could be adapted to embedded mobile applications. Meanwhile, the Convolutional Block Attention Module (CBAM) was embedded in the MobilenetV2 to enhance the feature extraction capability of the model. Furthermore, the Refinement Residual Blocks (RRBs) were introduced into the main branch and side branch of PSPNet to enhance the segmentation result. An algorithm to measure trunk diameter was proposed, which used the segmentation results to determine the trunk outline and the normal of the centerline. The Euclidean distance of the intersection point of the normal with the trunk profile was obtained and its average value was regarded as the final trunk diameter. Compared with the original PSPNet, the Intersection-over-Union (IoU) value, PA value and Fps of the improved model increased by 0.67%, 1.95% and 1.13, respectively, and the number of parameters was 5.00% of that of the original model. Compared with other segmentation networks, the improved model had fewer parameters and better segmentation results. Compared with the original network, the trunk diameter measurement algorithm proposed in this research reduced the average absolute error and the average relative error by 3.75 mm and 9.92%, respectively, and improved the average measurement accuracy by 9.92%. To sum up, the improved PSPNet jujube trunk segmentation algorithm and trunk diameter measurement algorithm can accurately segment and measure the diameter in the natural environment, which provides a theoretical basis and technical support for the clamping of jujube harvesting robots.

**Keywords:** red jujube tree; PSPNet; MobilenetV2; trunk segmentation; trunk diameter measurement

## 1. Introduction

Harvesting of ripe fruits is one of the important parts of fruit production and it is also the part with the largest labor demand and highest labor intensity [1]. However, with the development of social urbanization, a large number of people who used to work in agriculture have migrated to cities to engage in non-agricultural work. This has resulted in the lack of a sufficient agricultural labor force, which seriously affects agricultural production. Therefore, realizing the potential of mechanization and intelligence in the form of machines to engage in agricultural production, instead of human beings, effectively alleviates the labor shortage and promotes industrial development [2,3]. The extraction of trunk information plays an important role in research concerning fruit harvesting [4]. The planting pattern of existing orchard is regular, providing an ideal environment for

orchard robots to locate fruit trees [5]. However, there are light posts, water pipes, weeds and other sundries in orchards, which makes it more difficult for orchard robots. In the natural environment, it is difficult for picking robots to accurately distinguish different types of fruit trees, plan picking paths and measure the diameter because of the layout of the trunks and the irregular shape of branches [6,7]. These challenges to picking robots reduce harvesting efficiency. The emergence of big data and artificial intelligence has promoted the development of intelligent agriculture, and provided new ways of thinking to solve problems. Big data and artificial intelligence have promoted the development of intelligent agriculture and have provided new ideas to solve problems [8].

Tree trunk diameter measurement is an important step in the harvesting process of the vibration harvesting robot and the result of trunk segmentation directly affects the accuracy of trunk diameter measurement. Trunk segmentation often uses RGB images to extract the trunk, but, thanks to the development of hardware such as laser scanners and laser radars, many scholars have used the point cloud method to reconstruct and segment trunks [9–12]. However, this method can only extract one piece of trunk information at a time, and its detection range is small. The fusion of RGB images and point cloud data acquired by a laser scanner were used to improve the ability of trunk detection [13]. Nonetheless, this method requires high-quality point cloud data, and these point cloud data are easily segmented by light, which is easily lost in real scenes, affecting the reconstruction result. In addition, the laser scanner gets more redundant non-relay data, which makes it difficult to identify and match the correct relay data with the relay image. In contrast, the RGB or RGBD images of the method based on the use of a camera to obtain the trunk information are less affected by the environment, which is beneficial to the application of orchard picking robots. The method of obtaining trunk information based on the use of a camera first takes pictures of orchards with different cameras, then recognizes the trunk by its appearance, shape, texture, color and spatial relationship, and, finally, measures the trunk diameter by using the extracted trunk information [14–16]. The depth camera has been widely used in tree trunk diameter detection because it can obtain color images and depth images [15,16]. Machine learning and image processing methods are used to segment the trunk. However, it is necessary to find the model suitable for the data set by adjusting the threshold or gray histogram. Table 1 summarizes detailed tree trunk segmentation based on different data types.

**Table 1.** Summary of references regarding trunk segmentation methods based on different data types.

| Reference | Event | Data Type | Method | Result |
|-----------|-------|-----------|--------|--------|
| [11] | Hackenberg et al. established a high-precision tree segmentation method. | Terrestrial laser scan point clouds | A statistical method of cylinder radii was presented, based on point clouds data. | The total relative error was 8%. |
| [12] | Bargoti et al. presented a identification method of individual apple trees. | LiDAR point clouds data | Hidden Semi-Markov Model and Hough Transform was used to detect trunk, based on LiDAR data. | The accuracy of tree segmentation was 89%. |
| [13] | Shalal et al. presented a trees segmentation algorithm to discriminate between trees and non-tree objects | laser point clouds and camera images | A data fusion method of camera and laser scanner was proposed to detect trunk. | The detection accuracy was 96.64%. |
| [14] | Chen et al. presented A trunk detection algorithm based on multi-sensor integration technology | RGB images | HOG and SVM were used to train classifier, the gray histograms were used to optimize the classifier and Robert cross edge detector was used to improve accuracy. | The recall and accuracy of citrus trunk recognition experiments were 92.14% and 95.49%. |

**Table 1.** *Cont.*

| Reference | Event | Data Type | Method | Result |
|---|---|---|---|---|
| [15] | Shen Yue et al. proposed a fast tree trunk recognition method based on tree trunk features. | RGBD images | Super pixel segmentation was used for trunk segmentation and parallel edge feature detection was used to detect the trunk edge. | The recognition accuracy of trunk under normal illumination was 91.35%. |
| [16] | Liu Hui et al. proposed a fast trunk segmentation algorithm. | RGBD images | Super pixel algorithm was used to segment trunk and the color matching of the super pixel blocks was used to distinguish the trunk from the non-trunk. | The detection accuracy was 95. 0%. |

In recent years, the segmentation model based on deep learning has been widely used in production practice, and has achieved good results [17–19]. Convolutional Neural Network (CNN) has the ability to process high resolution image data and is widely used. Moreover, by sharing network weights among many convolution layers, a reasonable calculation time is realized and the detection efficiency is accelerated.

The application of deep learning in agriculture is mainly divided into two types: image semantic/instance segmentation and object detection. Object detection is widely used in fruit harvesting [20], and image semantic/instance segmentation is widely used in fruit quality detection and trunk detection [4,21,22]. Zhang et al. used an R-CNN-based object detection technology to detect the visible part of apple branches in a canopy, which was trained to capture tree structure [23]. Majeed et al. used the pre-trained SegNet framework to segment the trunk and branches from the background. The average BFScores of the trunk and branches were 0.93 and 0.88, respectively [24]. The research also conducted a dormant season with young one-year-old apple trees. Gao et al. reported using Faster R-CNN to detect various objects in apples, branches and trunks under the condition of whole leaves [25]. However, their work has not been optimized for detecting branches with different varieties to estimate the shaking position. In the natural environment, uneven illumination and various sundries, such as water pipes, have a great impact on the trunk segmentation of red jujubes.

In addition, jujube trees at different distances have a great influence on the segmentation effect of jujube trees in the same field of view. The main purpose of this research was to construct a trunk segmentation network of red jujubes suitable for mobile terminals by using deep learning networks, and to measure the trunk diameter of red jujubes. The research objectives were as follows:

(1) MobileNetV2 network was used as the backbone of the segmentation network to reduce the parameters and the model size.
(2) CBAM was introduced into the backbone network to improve the feature extraction ability of the network.
(3) RRB was introduced into the main branch and side branch to obtain more image details and realize accurate and efficient jujube tree segmentation.
(4) A measurement method to accurately measure trunk diameter was proposed.

## 2. Materials and Methods

In this research, images of jujube trees in complex field environment were taken as the research object. Trunk segmentation and diameter measurement experiments were carried out. The jujube tree image was collected in September 2021 at the date garden of No. 13 Company in Alar City, Xinjiang Uygur Autonomous Region.

### 2.1. Image Data Acquisition

Two kinds of jujube, namely, Junzao and Huizao, were selected as the collection objects. Intel RealSense D435i was used to obtain the RGB images and depth images of jujube trees.

In order to make the image background closer to the mechanical harvesting environment and increase the diversity of image samples, the distance between the camera lens and the trunk of the jujube tree was 50–100 cm during the process of image acquisition.

The image acquisition environment included sunny days and cloudy days, and included multi-targets, branches, leaves, sundries and other conditions. A total of 1038 images of red jujubes were collected, a few of which are shown in Figure 1.

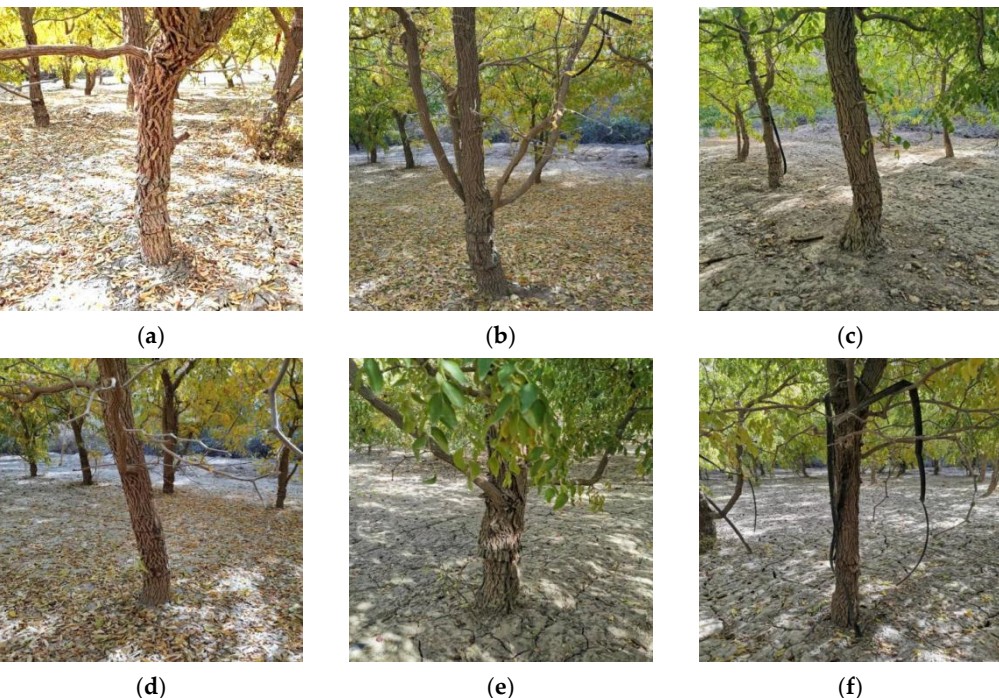

|     |     |     |
| --- | --- | --- |
| (**a**) | (**b**) | (**c**) |

| (**d**) | (**e**) | (**f**) |

**Figure 1.** Trunks of red jujubes in different scenes. (**a**) Sunny day, (**b**) Cloudy day, (**c**) Multiple targets, (**d**) Shading by branch, (**e**) Shading by leaves, (**f**) Shading by sundries.

The main instruments used in this research were Intel RealSense D435i, camera bracket, vernier caliper and a laptop. The depth camera was fixed on the camera bracket and connected with the laptop. The information collection equipment is shown in Figure 2a. Intel RealSense Viewer v2.50.0 was used to capture the image information and depth information of the trunk of red jujube, as shown in Figure 2b. Resolution of both RGB images and depth images was set to 640 × 480. As shown in Figure 2c, the trunk diameter was measured by using digital vernier calipers. Each tree was measured three times, and the average of the three trunk diameters was used as the final trunk diameter.

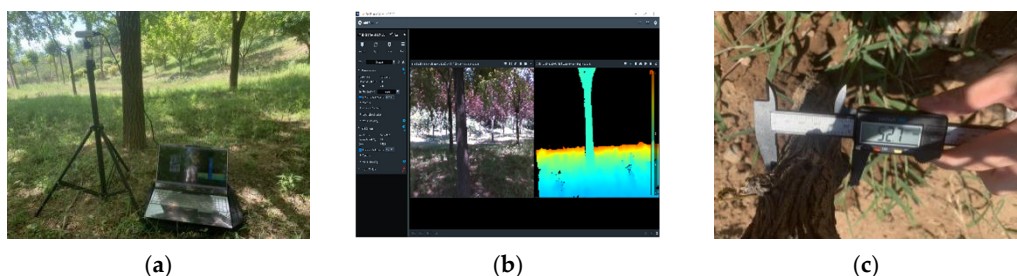

|     |     |     |
| --- | --- | --- |
| (**a**) | (**b**) | (**c**) |

**Figure 2.** Trunk information collection. (**a**) Information acquisition equipment, (**b**) Information acquisition interface, (**c**) Digital display vernier caliper measurement.

### 2.2. Data Annotation and Dataset Division

In this research, the images were cropped to 480 × 480, and the trunk below the first branch was selected as the main trunk of red jujube. Labelme was used as a point-by-point

labeling of the tree trunk, as shown in Figure 3. After labeling, 90% of the data set was divided into training set and validation set by 9:1, and the 10% was divided into the test set. The final number of image samples for training set, validation set and test set were 803, 110 and 113, respectively.

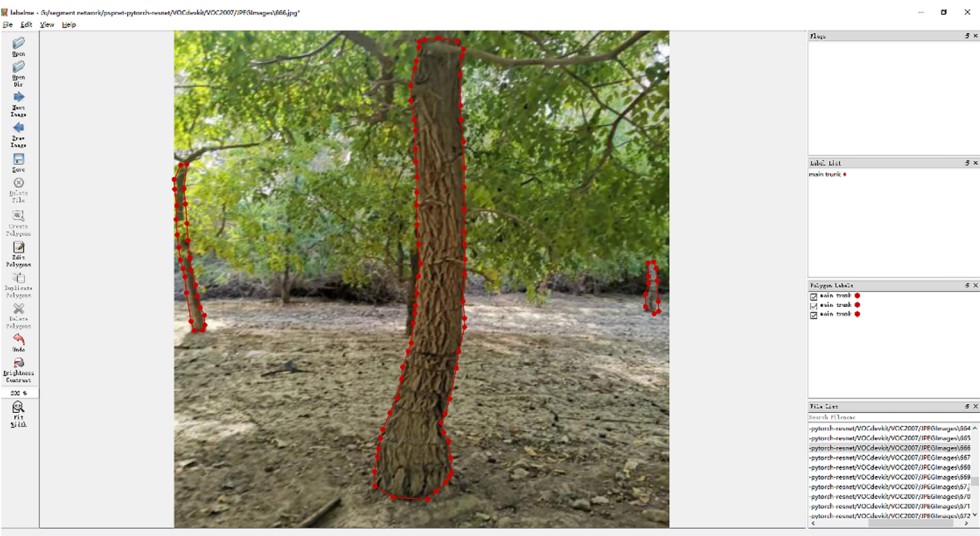

**Figure 3.** Labelme image annotation.

## 2.3. Improvement of PSPNet Segmentation Model

### 2.3.1. Baseline PSPNet Model

In a test on the dataset, the Fully Convolutional Network (FCN) misidentified a boat as a car due to the similarity of the shape features of the boat and the car. To enhance the scene perception capability of the network, and further improve the recognition of the network, PSPNet was proposed by Zhao in 2017 [26]. The original PSPNet mainly consisted of a backbone network (CNN) and Pyramid Pooling Module. The network structure diagram is shown in Figure 4.

As network depth increased, the network was able to extract complex features better, but there were also some side effects, such as gradient disappearance and network degradation. By adding a fast connection branch to ResNet, the problems were alleviated [27]. The original PSPNet used ResNet50 as the backbone network. The Block of ResNet50 contained three convolution layers: two $1 \times 1$ convolution layers and one $3 \times 3$ convolution layer. The three convolution layers are shown in Figure 5. In this block, the dimension of the feature map was subtly reduced or expanded by the $1 \times 1$ convolution layer, so that the number of filters in the $3 \times 3$ convolution layer was not affected by the input of the previous layer, and the next layer was affected. At first, the dimension of the middle $3 \times 3$ convolution layer was reduced by a $1 \times 1$ convolution layer, which reduced the amount of calculation, and then another $1 \times 1$ convolution layer was used to restore it. It not only maintained the accuracy of the model, but also reduced the network parameters and the amount of calculation, saving calculation time.

As the core component of PSPNet, the Pyramid pooling module divided the feature layer into several areas with different sizes. Each region independently generated different levels of feature maps, and finally the images were spliced, thus reducing the loss of information among different regions. The pyramid pooling module was a four-level module with pool cores of $1 \times 1$, $2 \times 2$, $3 \times 3$ and $6 \times 6$, respectively. The input feature map was globally pooled by a $1 \times 1$ pooling core through the pyramid pooling module and a single feature map was generated. The $2 \times 2$ pooling core divided the feature map into $2 \times 2$ sub-regions and pooled each sub-region. Similar to the $2 \times 2$ pooling cores, the $3 \times 3$ and the $6 \times 6$ pooling cores divided the input feature maps into $3 \times 3$ sub-regions and $6 \times 6$ sub-regions, respectively. Finally, the output feature map was expanded by linear

interpolation, so that it had the same size as the first input feature map and the final output feature map was obtained by means of concating.

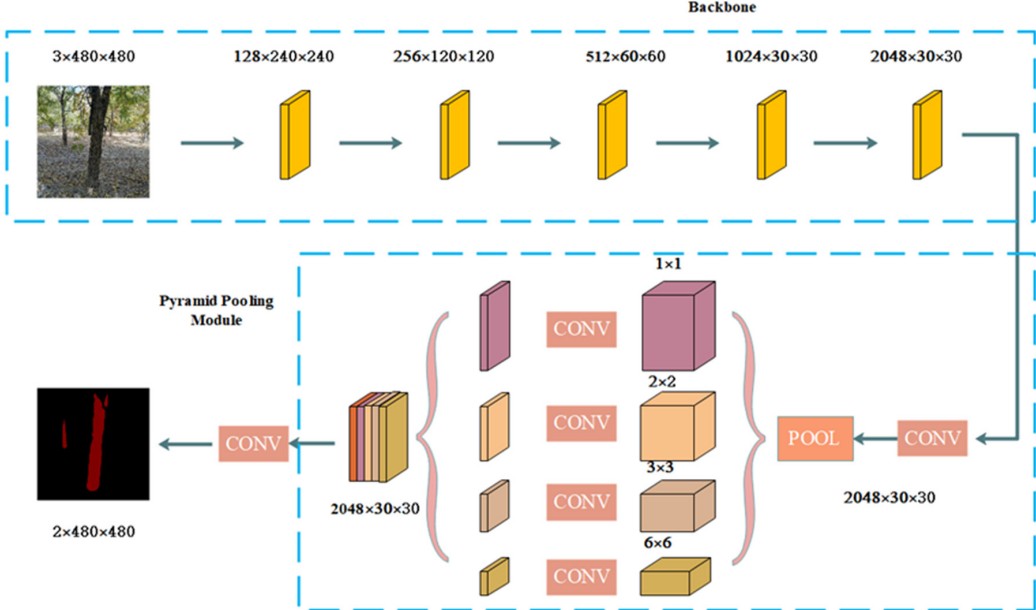

**Figure 4.** PSPNet network structure. Where: CONV represented the convolution operation of the feature map. POOL represented the pooling operation of the feature map.

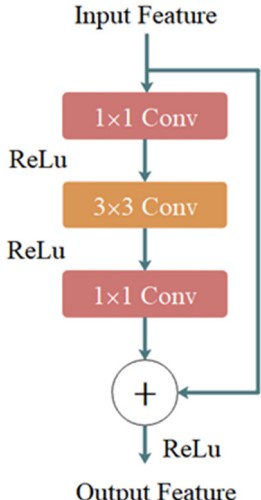

**Figure 5.** ResNet50 Block structure. Where: ReLu represents the rectified linear units. Conv represents the convolution layer.

According to the characteristics of the data set, the PSPNet was improved and optimized, and a semantic segmentation network of the jujube trunk, based on the improved PSPNet model, was constructed. The following three important improvements were proposed in the main feature extraction network and the enhanced feature extraction module, based on the original PSPNet network: (1) The light weight network MobileNetV2 was selected as the backbone network, (2) CBAM was introduced in the backbone of the improved PSPNet network and (3) RRB was introduced into the pyramid pooling module. The motivation and details of the above three improvements are summarized as follows.

2.3.2. Backbone Network Based on MobilenetV2

The main feature extraction network commonly used in the original PSPNet model was ResNet50. In the training process of the network model, a large number of convolution

calculations slowed down the update speed of the model parameter and the result affected the speed of feature extraction. In the computing environment of embedded devices, MobileNetV2 [28] was selected as the backbone network. Depth-wise separable convolution was used in MobileNetV1 [29]. Compared with the standard convolution, when the number of weight parameters was the same, the parameters of the model were greatly reduced and the calculation speed of the network was improved. MobileNetV2 put forward the inverted residual with linear bottleneck, which further reduced the memory occupation in the process of model reasoning, and was more suitable for embedded hardware design.

MobileNetV2 used depth-wise separable convolution instead of standard convolution to extract features from the feature map, as shown in Figure 6. Depth-wise separable convolution included depth convolution and point-by-point convolution. Depth convolution was a convolution kernel with a channel number, which was responsible for only one channel, through a convolution layer with a channel number and a convolution kernel size of $3 \times 3$, the input feature map was convoluted on a single channel of the feature map. Point-by-point convolution consisted of $1 \times 1 \times M$ convolution kernel. The feature maps of the previous step were weighted and combined in the depth direction to generate a new feature map. Compared with the traditional standard convolution, the parameters and operation cost of depth separable convolution were greatly reduced.

In the high-dimensional space, the nonlinear expression of features could be effectively increased by performing the operation of ReLU activation function on the convolution layer, but the feature information of the feature map would be lost when the operation of ReLU activation function was performed on the low-dimensional convolution layer. In order to prevent the loss of information, a linear bottleneck structure was proposed. When calculating the low-dimensional convolution layer, the ReLU activation function was not used for nonlinear transformation, thus reducing the information loss.

In the construction of a deep network structure, a bottleneck structure was put forward to alleviate the problems of gradient disappearance and network degradation. Compared with the standard convolution, MobileNetV2 had fewer channels because of the depth-wise separable convolution. It was bottleneck channeled and then the feature map was convoluted to extract features, resulting in the loss of feature information. Therefore, an Inverted Residual structure was proposed. The channel of the feature map was expanded by the convolution layer with the convolution kernel $1 \times 1$, then the feature was extracted by the deep convolution layer with the convolution kernel $3 \times 3$, and, finally, the channel was compressed by the point-by-point convolution with the convolution kernel $1 \times 1$. The specific parameters of MobileNetV2 network are shown in Table 2.

**Table 2.** Specific parameter list of mobilenetv2 network.

| Size of Input | Operators | Channel Dimension Expansion Factor | Channel Dimension | Stride |
|---|---|---|---|---|
| $3 \times 480 \times 480$ | Conv2d | - | 32 | 2 |
| $32 \times 240 \times 240$ | Bottleneck $\times$ 1 | 1 | 16 | 1 |
| $16 \times 240 \times 240$ | Bottleneck $\times$ 2 | 6 | 24 | 2 |
| $24 \times 240 \times 240$ | Bottleneck $\times$ 3 | 6 | 32 | 2 |
| $32 \times 120 \times 120$ | Bottleneck $\times$ 4 | 6 | 64 | 2 |
| $64 \times 60 \times 60$ | Bottleneck $\times$ 3 | 6 | 96 | 1 |
| $96 \times 60 \times 60$ | Bottleneck $\times$ 3 | 6 | 160 | 2 |
| $160 \times 30 \times 30$ | Bottleneck $\times$ 1 | 6 | 320 | 1 |
| $320 \times 30 \times 30$ | Conv2d | - | 1280 | 1 |
| $1280 \times 30 \times 30$ | Avgpool | - | - | - |
| $1280 \times 1 \times 1$ | Conv2d | - | 2 | 1 |

Where: Conv2d represented the convolution operation of the feature map with the convolution kernel $1 \times 1$. Bottleneck represented the feature map needing to undergo three convolution operations: one $1 \times 1$ convolution layer, one $3 \times 3$ convolution layer and one $1 \times 1$ convolution layer. Avgpool represented the average pooling operation of the feature map.

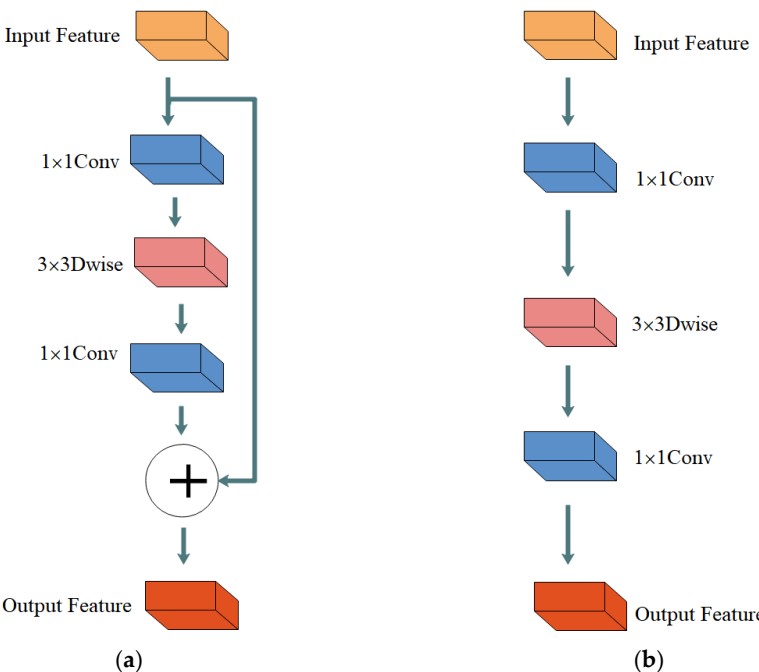

**Figure 6.** Inverted residual structure with linear bottleneck. (**a**) Stride was one. (**b**) Stride was two. Where: Conv represents convolution layer. Dwise represents the Depthwise convolution layer.

2.3.3. Backbone Feature Extraction Network Embedding CBAM

In order to extract more jujube trunk features in a complex environment, adding an attention mechanism to the segmented network was one of the methods to improve the network's ability to extract features [30]. The research object of this research was the trunk of red jujube, which had a high similarity with the surrounding environment. Therefore, CBAM was introduced into the improved trunk extraction network, and the feature extraction ability regarding the jujube trunk was improved.

In this research, MobileNetV2 was used as the backbone network of the model, and it contained different bottleneck structure modules. They were composed of the same depth separable convolution layers and the jump structure between the layers overcame the problems of generalization and gradient explosion in deep learning performance so as to learn the depth feature information better. In order to improve its special extraction capability, CBAM was introduced after the fifth layer and the seventh layers of the bottleneck structure.

Convolutional Block Attention Module (CBAM) [31] was put forward in 2018. It uses global average pooling and maximum pooling to enhance the feature information of the target region. This module can enhance image features over the channel, and improve the significance of important features. The module consists of two independent modules: Channel Attention Module (CAM), which pays attention to the channel, and Spatial Attention Module (SAM), which pays attention to space of the feature map. The formula for calculating the channel attention module is shown in Equation (1).

$$
\begin{aligned}
M_c(F) = \ & \sigma(MLP(AvgPool(F)) + MLP(MaxPool(F))) \\
= \ & \sigma\left(W_1\left(W_0\left(F_{avg}^c\right)\right) + W_1(W_0(F_{max}^c))\right)
\end{aligned}
\tag{1}
$$

where: $\sigma$ represents the activation function. *MLP* represents the multi-layer perceptron. $W_0$ and $W_1$ represent the weight parameter. The value $F_{avg}^c$ represents the average pooled feature map, $F_{max}^c$ represents the maximum pooled feature map, *AvgPool* represents the average pooling operation of the feature map *F*, and *MaxPool* represents the maximum pooling operation of the feature map *F*.

The formula for calculating the spatial attention module is shown in Equation (2).

$$M_s(F) = \sigma\big(f^{7\times7}([AvgPool(F); MaxPool(F)])\big)$$
$$= \sigma\big(f^{7\times7}\big(\big[F^s_{avg}; F^s_{\max}\big]\big)\big) \tag{2}$$

where: $\sigma$ is the activation function and $f^{7\times7}$ is the convolution operation with convolution kernel size of $7 \times 7$. The value $F^s_{avg}$ is the feature map, indicating average pooling and $F^s_{max}$ is the feature map, indicating maximum pooling. *AvgPool* represents the average pooling operation of the feature map $F$ and *MaxPool* representes the maximum pooling operation of the feature map $F$.

When the feature map was input to the module, the channel attention module acquired two 1-dimensional vectors from the input feature matrix through the global maximum pooling and global average pooling, and merged the vectors according to the channel dimension to generate the channel compression weight matrix. The channel compression weight matrix was multiplied with the input feature matrix to obtain the channel attention module and output feature maps. After the output feature mapping entered the spatial attention module, two 2-dimensional vectors were obtained by global maximum pooling and global average pooling, respectively. Two 2-dimensional vectors were convolved in the convolution layer by splicing to generate the weight matrix of spatial compression. The spatially compressed weight matrix was multiplied with the input feature mapping to obtain the output feature mapping of the spatial attention module. The CBAM structure is shown in Figure 7.

### 2.3.4. Improved PSPNet Model Embedding RRB

In order to improve the ability of the network to extract backbone features in a complex environment, Refinement Residual Block (RRB) was introduced into the main branch and the side branch of PSPNet [32]. The module was mainly composed of $1 \times 1$ and $3 \times 3$ convolution layers, and its network structure is shown in Figure 8. In this research, the RRB module was used in the main branch and side branch of the improved PSPNet, which resulted in more detailed feature information and enhanced the feature recognition capability.

### 2.3.5. Improved PSPNet Model

Figure 9 shows the improved PSPNet used for red jujube trunks in natural environments. The model structure was mainly composed of a backbone feature extraction network and a pyramid pooling module. The former used MobileNetV2 as the backbone network of the model, which reduced the model scale and improved the accuracy of the model at the same time. The CBAM was introduced in the backbone feature extraction network to suppress useless feature extraction, and, at the same time, enhanced the attention to important features. RRB was introduced in the main branch and side branch of the model, so as to obtain more detailed feature information and enhance the feature recognition capability.

### 2.3.6. Measurement Method of Jujube Tree Diameter Based on Centerline

In the vibration harvesting of jujube trees, the mobile equipment equipped with trunk measurement often moves to 50 cm~100 cm of jujube trees to identify and measure the trunk of jujube trees. In this research, the trunk segmentation network was used to segment the trunk of fruit trees from the complex environment. Through image processing, the center line of the trunk was detected, and then the normal of each point on the center line was found. The coordinates of the intersection point were obtained by the intersection point of normal lines and boundary lines. By means of the Euclidean distance formula the distance between two points was found. This average distance was used as the pixel distance of the final trunk. Finally, according to the relationship between the camera and the image, the diameter of the measured trunk was obtained. The flow chart of the jujube tree diameter measurement method, based on the centerline, is shown in Figure 10.

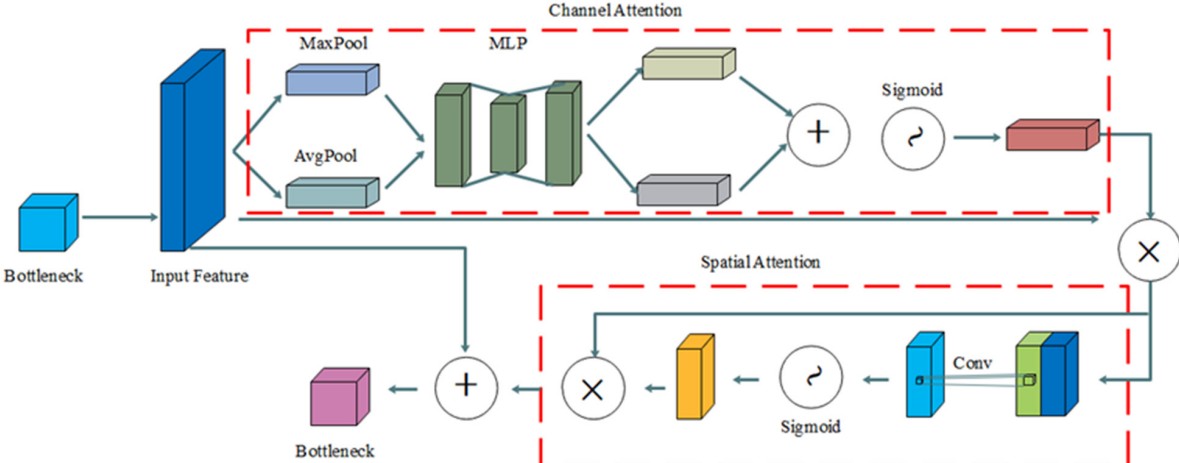

**Figure 7.** CBAM structure. Where: Conv represents convolution layer. MLP represents multilayer perceptron. *AvgPool* represents the average pooling operation of the feature map. *MaxPool* represents the maximum pooling operation of the feature map.

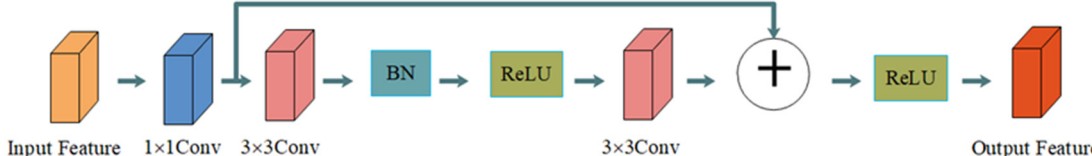

**Figure 8.** RRB structure. Where: Conv represents convolution layer. ReLU represents the rectified linear units. BN represents the batch normalization.

The normal of center line was calculated. To extract the trunk centerline of the jujube tree, the trunk image was extracted by dividing the network and could be binarized by the OTSU method, and then the trunk centerline could be extracted by skeletonizing in the scikit-image library, as shown in Figure 11d. Finally, the normal of each point on the center line was calculated according to the tangent of each point $(x_i, y_i)$ on the center line. The normal calculation method is shown in Formula (3).

$$y = -\frac{1}{k}x + \frac{1}{k}x_i + y_i \tag{3}$$

where: $k$ represents the tangent slope of the passing point $(x_i, y_i)$, $x$ represents the abscissa of the point on normal line of the trunk centerline ajnd $y$ represents the ordinate of the point on the normal line of the trunk centerline.

The center of the line consisted of two types of points: the end point and other points on the center line. The tangent values of the two points could be obtained using the method described below.

Method for calculating the tangent at the end point: if P $(x_p, y_p)$ was an end point on the center line, there was only one point Q $(x_q, y_q)$ in the eight neighborhoods of this point, as shown in Figure 12, and the tangent line at the end point was the tangent line $y_{PQ}$ of the line segment PQ, and the tangent of this point was shown in Equation (4).

$$y_{PQ} = \frac{y_p - y_q}{x_p - x_q}x - \frac{(y_p - y_q)x_q}{x_p - x_q} + y_q \tag{4}$$

where: $y_{PQ}$ was the tangent of line segment PQ.

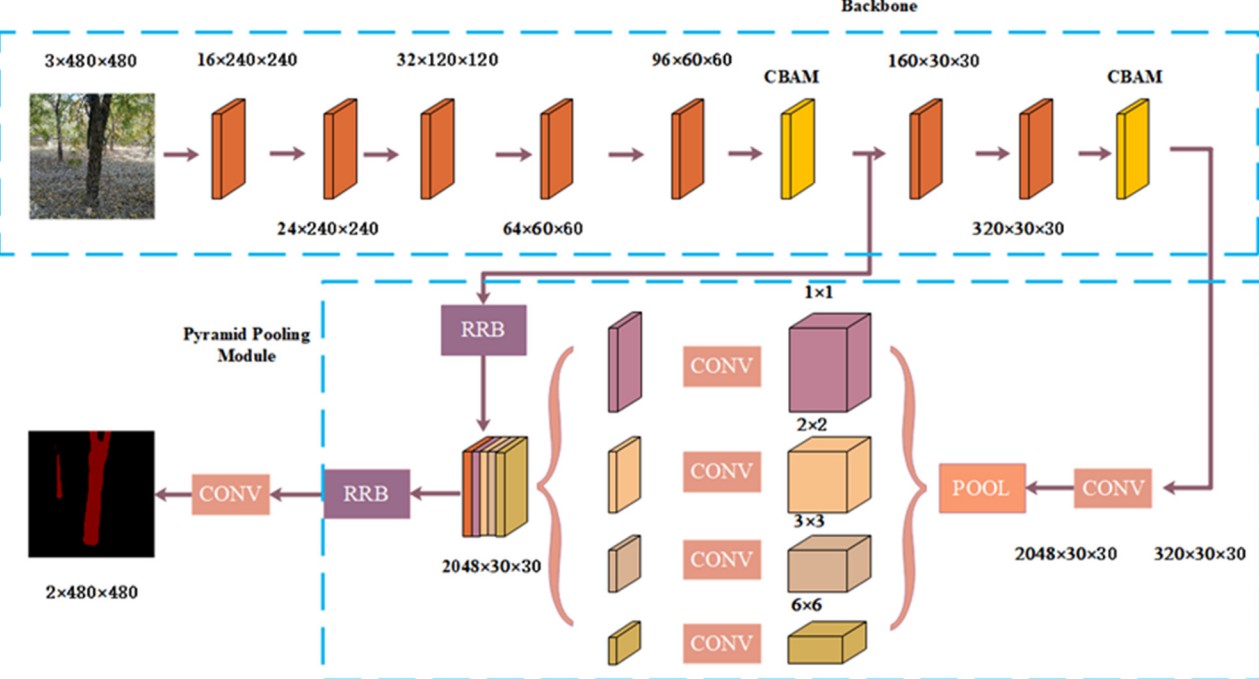

**Figure 9.** Structure diagram of jujube trunk segmentation model based on improved PSPNet. Where: CONV represents the convolution operation of the feature map. POOL represents the pooling operation of the feature map. RRB represents the refinement residual block.

**Figure 10.** Flow chart of jujube tree diameter measurement method, based on centerline.

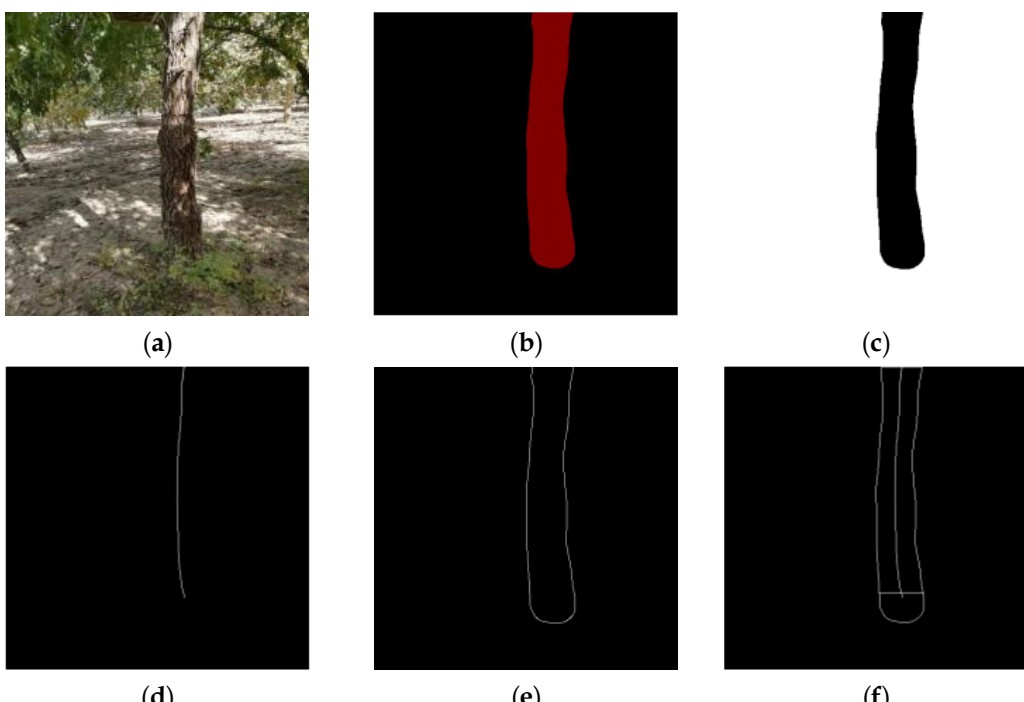

**Figure 11.** Trunk diameter measurement. (**a**) The original image of red jujube trunk. (**b**) The image resulting from the improved trunk segmentation algorithm. (**c**) The image resulting from image preprocessing. (**d**)The image resulting from centerline extraction. (**e**) The image resulting from edge extraction. (**f**) The image resulting from Intersection point calculation.

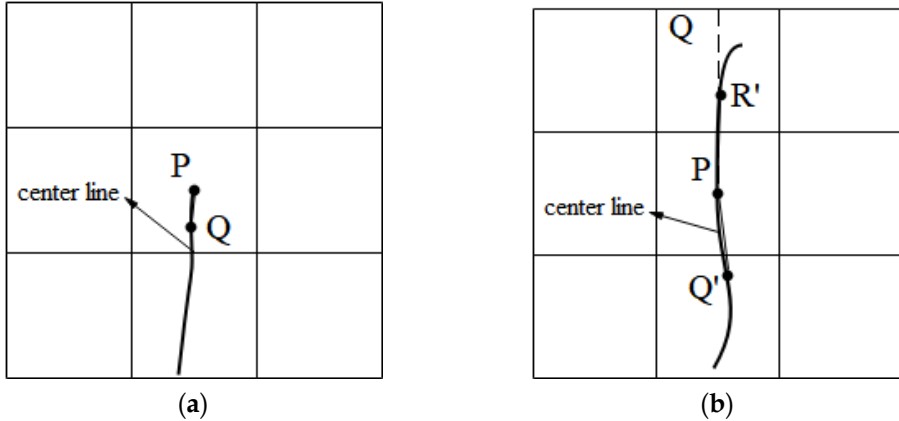

**Figure 12.** Tangent line at the center line. Where: (**a**) tangent line at end point. (**b**) tangent line at the center point on the line.

Calculation method of tangents at other points on the center line: if $P'(x_{p'}, y_{p'})$ was any point on the center line, there were two points $Q'(x_{q'}, y_{q'})$ and $R'(x_{r'}, y_{r'})$ in the eight neighborhoods of that point, and the tangent of that point was the direction of the midline of the two tangents. The tangent of this point was shown in Equations (5)–(7).

$$y_{P'Q'} = k_{P'Q'}x - k_{P'Q'}x_{p'} + y_{p'} \tag{5}$$

$$y_{P'R'} = k_{P'R'}x - k_{P'R'}x_{p'} + y_{p'} \tag{6}$$

$$y = k_{P'}x - k_{P'}x_{p'} + y_{p'} \tag{7}$$

where: $k_{P'Q'} = \frac{y_{q'} - y_{p'}}{x_{q'} - x_{p'}}$, $k_{P'R'} = \frac{y_{r'} - y_{p'}}{x_{r'} - x_{p'}}$, $k_{P'} = \frac{k_{P'Q'} + k_{P'R'}}{2}$. $y_{P'Q'}$ was the tangent of a line segment $P'Q'$; $y_{P'R'}$ was the tangent of a line segment $P'R'$; $y$ was the tangent of the point $P'$.

The distance between the pixels with diameter was calculated. After calculating the normal of each point of the center line, the trunk boundary line was detected by HoughLinesP in OpenCV. The distance between the intersection of the normal and trunk boundary lines was the pixel distance of trunk diameter at that point. O $(x_o, y_o)$ was any point on the center line. P $(x_p, y_p)$ and Q $(x_q, y_q)$ were the points on the tangent of the center line. A $(x_a, y_a)$ and B $(x_b, y_b)$ were the endpoints on the left boundary of the trunk. C $(x_c, y_c)$ and D $(x_d, y_d)$ were the endpoints on the right boundary of the trunk. The intersection of the normal of this point and the trunk boundary was M $(x_m, y_m)$ and N $(x_n, y_n)$. The Euclidean distance of the line segment MN was the diameter pixel distance. The distance of MN was shown in Equation (8).

$$d = \sqrt{(x_m - x_n)^2 + (y_m - y_n)^2} \tag{8}$$

where: $d$ was the distance of MN.

Trunk width detection. In this research, Intel RealSense D435i was used to collect the image of jujube trunk. The parameters of the camera are shown in Table 3.

**Table 3.** Specific parameter list of Intel RealSense D435i.

| Project | Parameter |
|---|---|
| depth field of view (FOV) | 85.2° × 58° × 94° |
| maximum output resolution | 1280 × 720 |
| minimum depth distance (m) | 0.1 |
| RGB sensor FOV (Before calibration) | 69.4° × 42.5° × 77° |
| RGB sensor FOV (After calibration) | 53.4° × 42.5° |

According to the parameters in the table, the angle β between adjacent pixels and the camera is shown in Equation (9). When measuring the trunk, the cross section of the trunk was regarded as a standard circle in this research, and the camera scanning of the trunk is shown in Figure 13.

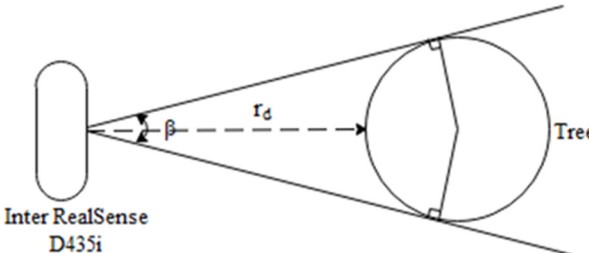

**Figure 13.** Schematic diagram of scanning trunk by camera.

The diameter of the trunk could be obtained from the following equation, and the trunk diameter from in Equation (10).

$$\beta = \frac{53.4°}{c}(n-1)(1+n_0) \tag{9}$$

$$D = 2r_d \frac{\sin(\beta/2)}{1 - \sin(\beta/2)} \tag{10}$$

where: $c$ was the number of columns in the depth image; $n_0$ was the number of invalid pixels between the calculated valid pixels and the previous valid pixels; $n$ was the number of pixel points on the arc; $\beta$ was the angle at which the camera scans the trunk section; $r_d$ was the distance from the camera to the trunk; $D$ was the diameter of the trunk.

## 3. Results and Discussion

This section first introduces the hardware and software used in the jujube tree trunk segmentation experiment and the process of data set training, then introduces the evaluation indices used in the experiment, then analyzes the influence of the improved parts on the experiment, and finally measures the trunk diameter with this model.

### 3.1. Experimental Platform and Model Training

The model proposed in this research was based on the improved PSPNet red jujube trunk segmentation and diameter measurement model, coded in Python and tested by the Pytorch deep learning framework. The test environment and hardware are shown in Table 4.

**Table 4.** Experimental environment.

| Configuration | Parameter |
|---|---|
| CPU | Intel(R) Core (TM) i7-10700K |
| GPU | NVIDIA GeForce RTX 3070 |
| Accelerated environment | CUDA11.1 CUDNN8.2.1 |
| Development environment | Pycharm 2021.3.2 |
| Operating system | Ubuntu 18.04 |

Setting of network super parameters. In the test model, the batch size was 2, the down sample factor was 16, the weight initial learning rate was 0.01, and the attenuation coefficient was 0.01. SGD was selected as the model optimizer, where momentum was 0.9 and the weight attenuation coefficient was 0.0001 [33]. The total number of iterations of training was 200 iterations. A model weight was saved in each iteration, and the model with the highest accuracy was selected as the final model. The change of the training set loss value with the number of iterations is shown in Figure 14. It can be seen from the figure that in the range of 0-50 iterations, the loss value of the improved model gradually decreased with increase of iterations, and the loss value of the model decreased fastest in this range. When the number of iterations was in the range of 50-200 iterations, the loss value of the model changed little and gradually tended to be stable and the network reached the convergence state.

### 3.2. Metrics for Model Performance Evaluation

In this research, the trunk of the jujube tree was regarded as one category, and the background as another category. In order to measure the effect of the model on jujube trunk segmentation, pixel accuracy (PA), IoU, Fps, network parameters, diameter relative error (*P*) and diameter measurement accuracy (*E*) were used as evaluation indices of model performance, and their calculation formulae were shown in Equations (11)–(14). Pixel Accuracy (PA), IoU, Fps and network parameters were used as evaluation indices of jujube tree trunk segmentation model. The higher the index value was, the more effective the model was. Relative diameter error and diameter measurement accuracy were the evaluation indices of trunk diameter measurement. The smaller the relative diameter error was, the more effective the model was, while the diameter measurement accuracy behaved opposite.

$$PA = \sum_{i=0}^{k} \frac{p_{ii}}{\sum_{j=0}^{k} p_{ji}} \times 100\% \qquad (11)$$

$$IoU = \sum_{i=0}^{k} \frac{p_{ii}}{\sum_{j=0}^{k} p_{ij} + \sum_{j=0}^{k} p_{ji} - p_{ii}} \times 100\% \qquad (12)$$

$$E = \frac{|C - V|}{C} \times 100\% \qquad (13)$$

$$P = 1 - E \tag{14}$$

where: $p_{ii}$ was the pixel point, the recognition result of which in class $i$ was class $i$; $p_{ij}$ was the pixel point, the recognition result of which in class $i$ was class $j$; $p_{ji}$ was the pixel point, the recognition result of which in class $j$ was class $i$; $k$ was the number of different categories of data sets, $k = 2$; $E$ was the relative error of the diameter of the trunk of jujube; $C$ was the measured value of jujube diameter measurement algorithm; $V$ was vernier caliper measurement of the trunk; $P$ was the measurement accuracy of trunk diameter.

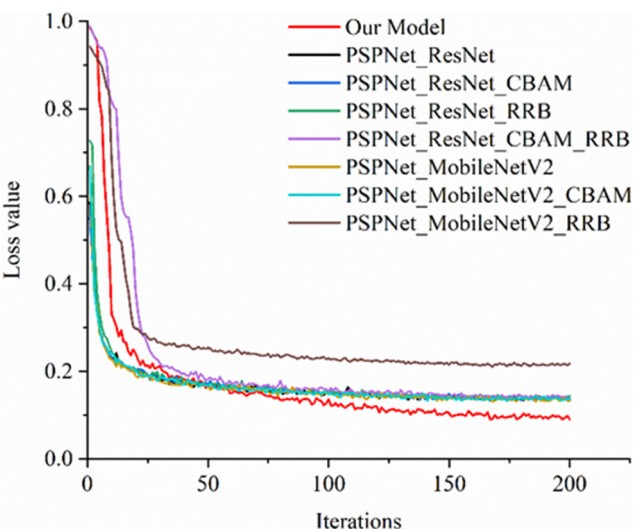

**Figure 14.** Changes of training set loss value of the model with iteration times.

### 3.3. Experimental Results and Analysis of Trunk Segmentation

In order to verify the trunk segmentation performance of the improved model, based on PSPNet, different structures of PSPNet models were tested on the jujube trunk data sets in a natural environment, and the effectiveness of different structures on the segmentation model performance were verified. The result is shown in Table 5.

As a classic segmentation network model, PSPNet can effectively segment the target in complex environments, but it is difficult to segment jujube trunks with limited computation. In the original PSPNet, ResNet50 was used as the main extraction network. The IoU and PA of the jujube trunk segmentation model were 81.21% and 89.44%, respectively, in the original PSPNet. Adding CBAM in the 13th convolution residual layer, meant the improved model increased the parameters and reduced the IoU of the model, used for trunk segmentation. Introducing RRB into the main branch and side branch of the network, the parameters of the improved model were basically unchanged, but the IOU improved. Introducing CBAM and RRB at the same time, resulted in improved IoU, compared with the original PSPNet, but the parameters of the two models were also increased. This could not meet the demand of embedded mobile devices. Compared with the original PSPNet, the IoU, PA and Fps of the jujube trunk segmentation model increased by 0.15%, 0.53% and 54.02, respectively, and the model parameter decreased by 20 times when MobileNetV2 was used as the backbone of the extraction network. Compared with the original PSPNet, the improved PSPNet, which took MobileNetV2 as the backbone extraction network and introduced the CBAM network after the 5th and 7th bottleneck structures, had some improvement in IoU, which verified the effectiveness of introducing the CBAM network. Compared with the original PSPNet and the improved PSPNet based on MobileNetV2, the improved PSPNet, which took MobileNetV2 as the backbone to extract the network and introduced RRB into the main branch and side branch of the network, increased by 0.61% and 0.46% in the IoU, respectively. PA increased by 0.95% and 0.42%, and the Fps increased by 5.07 and 0.82, respectively. Compared with the other algorithms based on PSPNet, the improved PSPNet, which took MobileNetV2 as the backbone network and introduced the

CBAM and RRB, was 0.67% higher in IoU, 1.95% higher in PA and 1.13 higher in Fps, and the algorithms parameter was only 5.00% of the original PSPNet.

**Table 5.** Comparison results of each module on model performance.

| Model | IoU/% | PA/% | Fps | Parameters |
|---|---|---|---|---|
| ResNet50 | 81.21 | 89.44 | 49.77 | $4.91 \times 10^7$ |
| ResNet50 + CBAM | 80.82 | 89.44 | 48.57 | $5.04 \times 10^7$ |
| ResNet50 + RRB | 81.41 | 90.86 | 51.02 | $4.91 \times 10^7$ |
| ResNet50 + CBAM + RRB | 81.80 | 90.56 | 48.76 | $5.04 \times 10^7$ |
| MobileNetV2 | 81.36 | 89.97 | 54.02 | $2.45 \times 10^6$ |
| MobileNetV2 + CBAM | 81.37 | 89.13 | 51.76 | $2.48 \times 10^6$ |
| MobileNetV2 + RRB | 81.82 | 90.39 | 54.84 | $2.45 \times 10^6$ |
| Ours model | 81.88 | 91.39 | 50.90 | $2.48 \times 10^6$ |

The Jujube trunk data set contained single target, multi-target, sundries, occlusion and non-occlusion, etc. Compared with the label image, the original PSPNet had segmentation defects when segmenting the trunk with branches, as shown in Image 1 of Figure 15, while the improved PSPNet could better segment the trunk. When segmenting the trunk with leaves, the original PSPNet easily caused segmentation defects for the occluded part, as shown in Image 2 of Figure 15. The main reason was that the network mistakenly segmented this part as the background due to the characteristics of leaves and trunks being so different. The improved PSPNet enhanced the ability of feature extraction, and could segment the trunk with leaves better than the original PSPNet. Compared with the original PSPNet, the improved model was more accurate and complete in multi-target segmentation, as shown in Image 3 and Image 4 in Figure 15, which further verified the effectiveness of the improved model in jujube trunk segmentation. When segmenting the trunk of jujube with sundries, PSPNet could effectively identify the sundries as the background, so as to avoid the influence of sundries on the segmentation results, as shown in Image 2 and Image 4 in Figure 15.

Image 1 was a single-target jujube tree trunk image with branches. Image 2 was a single-target jujube tree trunk image with leaves and sundries. Image 3 was a multi-target jujube tree trunk image. Image 4 was a multi-target jujube tree trunk image with sundries, in which the yellow box was manually marked.

*3.4. Different Model Segmentation Results and Analysis*

In this research, five commonly used segmentation networks were used to test the data set of jujube tree trunks, and the test results are shown in Table 6. As can be seen from the table, the IoU value and PA value of the improved model were the highest among the experimental models, reaching 81.88% and 91.39%, respectively. The results show that the segmentation effect of the improved PSPNet model was better than other models. Compared with other models, the parameters of the model were the least among the experimental models, so the improved model had more advantages when it was applied to embedded mobile devices. However, the Fps value of the improved model was 50.90. The detection speed of this model was slightly lower than that of BiseNet, FCN and Unet, but better than Unet++ and DeepLab v3+. Therefore, the model could satisfy real-time detection of jujube trunks.

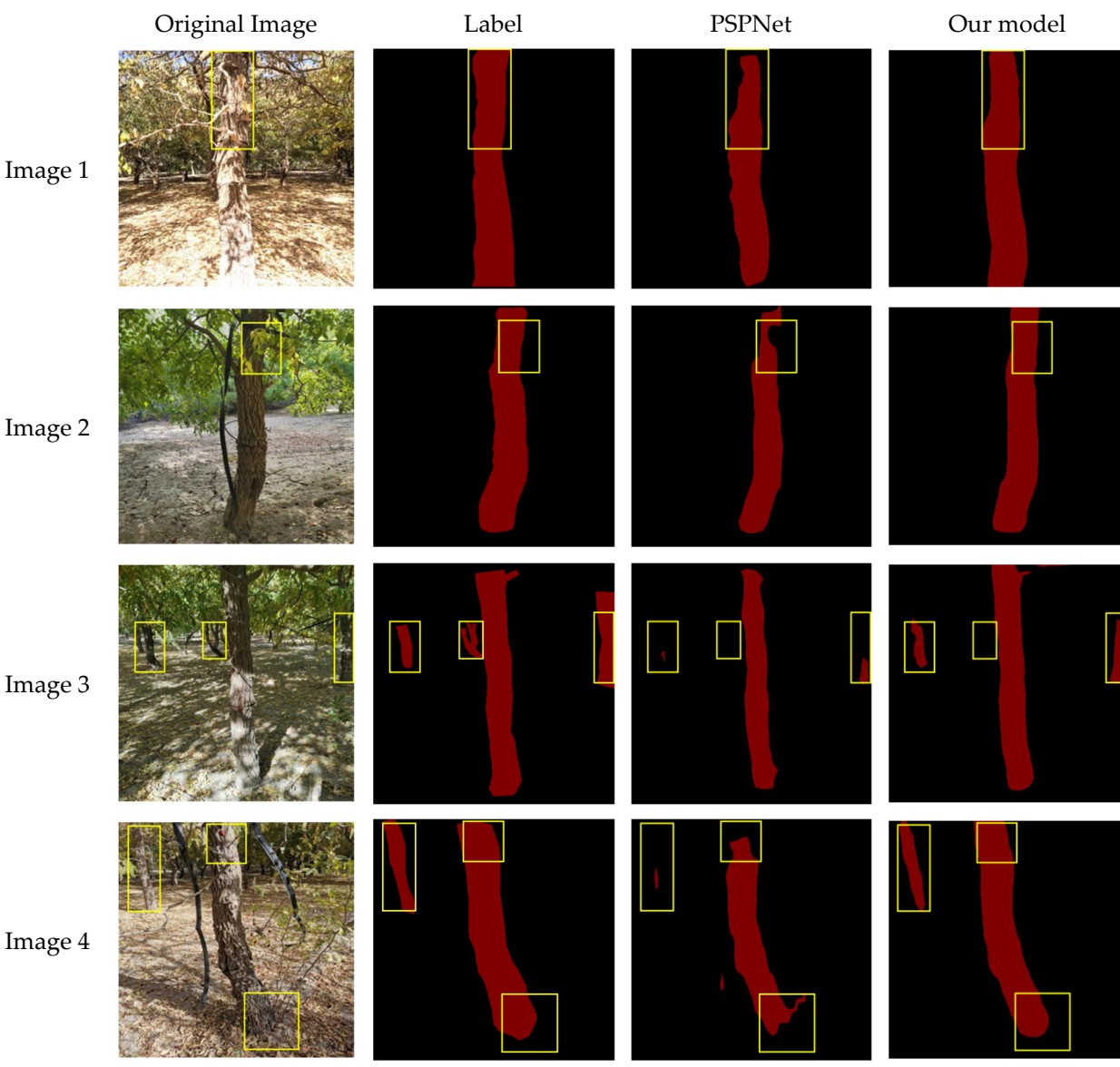

**Figure 15.** Results of jujube recognition by different algorithms.

**Table 6.** Comparison of test results of different segmentation networks on data sets.

| Model | IoU/% | PA/% | Fps | Parameters |
|---|---|---|---|---|
| BiseNet | 74.86 | 80.74 | 122.58 | $2.31 \times 10^7$ |
| DeepLab v3+ | 71.22 | 77.87 | 37.96 | $5.46 \times 10^7$ |
| FCN | 79.31 | 88.15 | 55.92 | $2.01 \times 10^7$ |
| Unet | 78.54 | 86.91 | 56.44 | $7.77 \times 10^6$ |
| Unet++ | 78.99 | 87.13 | 21.29 | $9.16 \times 10^6$ |
| Ours model | 81.88 | 91.39 | 50.90 | $2.48 \times 10^6$ |

Figure 16 shows the partial segmentation results of five common segmentation networks in the experiment regarding the jujube trunk data set. The improved PSPNet network and other experimental networks could complete the task of segmenting the trunk of jujube in different scenarios, such as on sunny or cloudy days, and could cope with multi-targets, tree branch occlusion, leaf occlusion and sundries. Compared with the other segmentation networks, the improved PSPNet network was superior to other experimental segmentation models in the segmentation of jujube trunks in different scenes, but there were still some problems of missing segmentation and wrong segmentation, as shown in Figure 16. As

an advanced segmentation network, DeepLab v3+ had a poor segmentation effect when segmenting the jujube trunk, as shown in DeepLab v3+ in Figure 16. The result of tree trunk segmentation by the network in the figure was not complete, and there were large areas of tree trunk missing and wrongly imaged. As a relatively new network, Bisenet missed points in different scenes. In the scene containing sundries, Bisenet mistakenly classified sundries into jujube trunks, as shown in Bisenet of Figure 16. FCN, Unet and Unet++, as the classical networks in segmentation tasks, had good segmentation performance, but there were also mistakes and omissions in the segmentation of jujube trunks in different scenes. On cloudy days, FCN, Unet and Unet++ mistakenly classified the background into jujube trunks, such as FCN, Unet and Unet++ of Figure 16. Compared with the other two kinds of segmentation networks, Unet had a poor segmentation result and there were many missing points in the scene covered by trees. In the environment containing clutter, all three segmentation networks segmented the clutter into red jujube trunks. When segmenting multi-targets, the three networks easily missed points for smaller and larger segmentation targets. However, in the scene of shading leaves, the three networks had better segmentation effects. The improved PSPNet network incorporated more feature extraction modules, so it had a strong segmentation effect, as shown in our model of Figure 16. Although there were some mistakes and omissions, the overall segmentation result was still better than other segmentation networks in the experiment.

### 3.5. Diameter Detection Results Based on Improved PSPNet

The diameter measurement method was proposed in this research. This method used the segmentation network to segment the trunk from the complex background, extracted the centerline of the trunk and calculated the normal of each point on the centerline. Then, the trunk edge was extracted and the intersection of the trunk edge and the normal was calculated. Finally, the Euclidean distance formula was used to calculate the distance between two points, and the average of the distances was calculated as the final trunk diameter. In this research, the diameter of 10 jujube trees was measured and the measurement results are shown in Table 7.

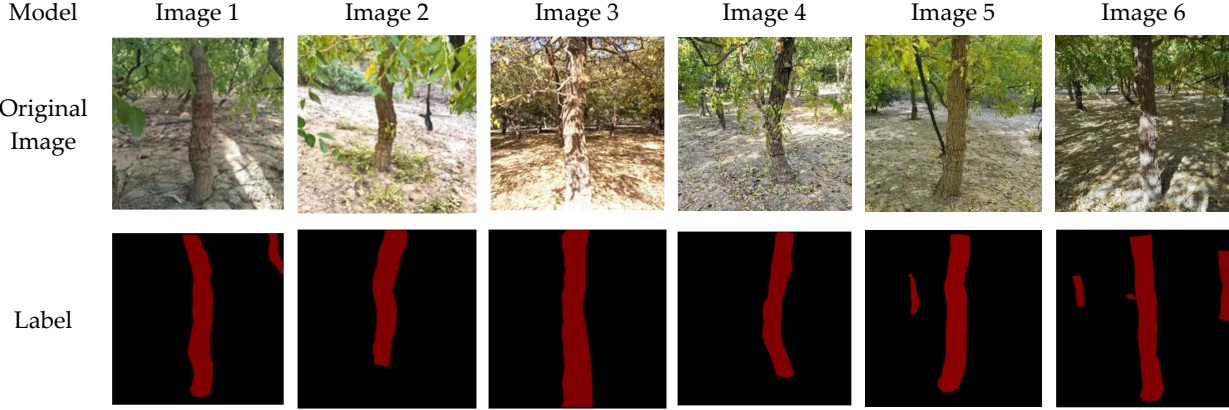

**Figure 16.** *Cont.*

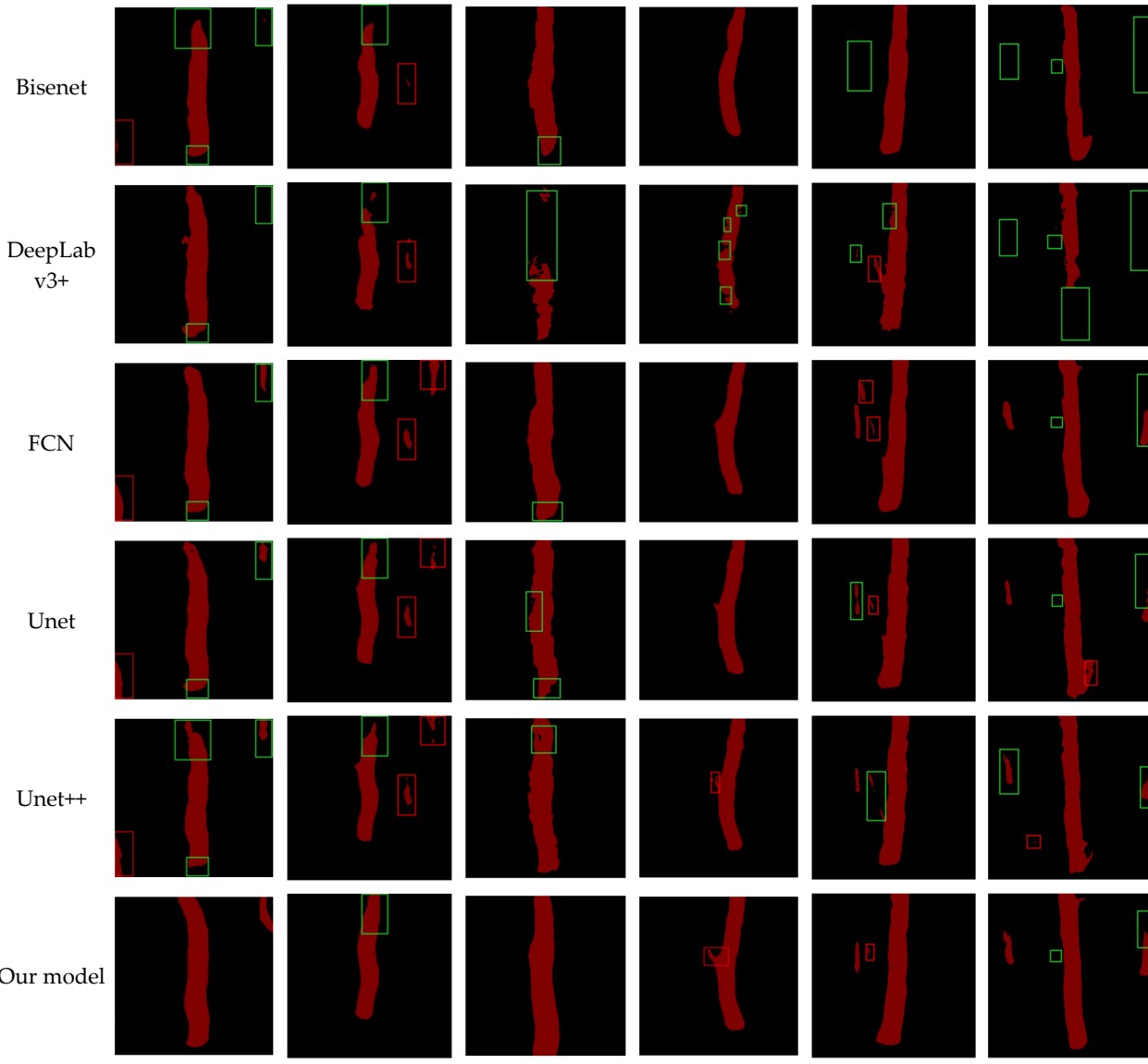

**Figure 16.** Segmentation results of jujube trunk by different segmentation models. Where: the red boxed area was mistakenly segmented. The green boxed area was incompletely segmented.

**Table 7.** Experimental environment.

| m_dis /mm | t_dia /mm | PSPNet | | | | | Improved PSPNet | | | | |
|---|---|---|---|---|---|---|---|---|---|---|---|
| | | p_dis /Pixel | m_dia /mm | a_err /mm | r_err /% | m_acc /% | p_dis /Pixel | m_dia /mm | a_err /mm | r_err /% | m_acc /% |
| 450.32 | 32.86 | 44.08 | 29.86 | 3.00 | 9.13 | 90.87 | 51.52 | 35.09 | 2.23 | 6.80 | 93.20 |
| 480.02 | 33.23 | 43.05 | 31.06 | 2.17 | 6.52 | 93.48 | 47.50 | 34.39 | 1.16 | 3.48 | 96.52 |
| 480.36 | 33.23 | 38.23 | 27.50 | 5.73 | 17.24 | 82.76 | 47.43 | 34.36 | 1.13 | 3.40 | 96.60 |
| 500.52 | 35.32 | 39.54 | 29.67 | 5.65 | 15.99 | 84.01 | 46.58 | 35.14 | 0.18 | 0.52 | 99.48 |
| 500.65 | 23.09 | 26.16 | 19.44 | 3.65 | 15.80 | 84.20 | 32.26 | 24.08 | 0.99 | 4.30 | 95.70 |
| 510.36 | 38.98 | 38.77 | 29.65 | 9.33 | 23.95 | 76.05 | 48.16 | 37.09 | 1.89 | 4.86 | 95.14 |
| 511.32 | 42.23 | 47.43 | 36.58 | 5.65 | 13.39 | 86.61 | 52.00 | 40.23 | 2.00 | 4.73 | 95.27 |
| 514.56 | 31.25 | 37.94 | 29.24 | 2.01 | 6.44 | 93.56 | 39.20 | 30.23 | 1.02 | 3.26 | 96.74 |
| 520.23 | 32.13 | 37.63 | 29.31 | 2.82 | 8.78 | 91.22 | 42.64 | 33.33 | 1.20 | 3.74 | 96.26 |
| 520.53 | 54.23 | 55.81 | 44.08 | 10.15 | 18.71 | 81.29 | 66.96 | 53.34 | 0.89 | 1.65 | 98.35 |
| average value | | | | 5.02 | 13.59 | 86.41 | - | - | 1.27 | 3.67 | 96.33 |

Where: m_dis was measuring distance, t_dia was true diameter, p_dis was pixel distance, m_dia was measuring diameter, a_err was absolute error, r_err was relative error and m_acc was measuring accuracy.

Compared with the actual diameter, the average absolute error of the trunk diameter measured by the original PSPNet was 5.02 mm. The average relative error was 13.59%. The average measurement accuracy was 86.41%. However, the accuracy of the trunk diameters measured by the improved PSPNet was above 90%. The average absolute error of the trunk diameter measured by the improved PSPNet was 1.27 mm, the average relative error was 3.67% and the average measurement accuracy was 96.33%. Compared with the original network, the average absolute error and average relative error of the improve PSPNet reduced by 3.75 mm and 9.92%, respectively, and the average measurement accuracy increased by 9.92%. The improved network can meet the requirements of forest resources investigation. The improved segmentation network, combined with the diameter measurement algorithm, can measure the diameter of the trunk well, and the average measurement accuracy of the trunk was 96.33%.

### 4. Conclusions

In the field environment, trunk segmentation and diameter measurement are greatly influenced by the orchard environment, such as sunny and cloudy, multi-target, tree branches, leaves, sundries and other orchard environments. An improved PSPNet network algorithm and a trunk diameter measurement algorithm were proposed to realize trunk segmentation and diameter measurement. Firstly, the improved trunk segmentation algorithm of the jujube tree, based on PSPNet, was improved. MobilenetV2 was selected as the backbone extraction network of PSPNet in this research, and CBAM was introduced after the fifth and the seventh layers of the bottleneck structure to improve the feature extraction capability of the model. RRB was introduced in the main branch and side branch of the model to obtain more detailed feature information and to enhance the feature recognition capability. Then, a trunk diameter measurement algorithm, based on image processing technology, was proposed. The algorithm used the segmentation result of the red jujube trunk segmentation model, based on the improved PSPNet, to calculate the trunk contour and the normal of the centerline, and, then, calculated the intersection of the normal and the trunk contour. The Euclidean distance of the intersection point was taken as its average value as the final trunk diameter result. The experimental results of the improved PSPNet jujube trunk segmentation algorithm showed that the IoU value, PA value and Fps value were 81.88%, 91.39% and 50.9, respectively and the parameter was only $2.48 \times 10^6$. The results of a trunk diameter measurement method showed that the average absolute error and average relative error were reduced to 1.27 and 3.67, respectively, and the average accuracy rate reached 96.33%. Therefore, the measurement result was good, meeting actual needs.

In summary, a diameter measurement method for the truck of red jujubes, based on improved PSPNet, was proposed in this research, and the feasibility and effectiveness of the measurement method were verified by experiments. Future work on the measurement method of red jujube trunk diameter will be as follows:

(1) Expand the types of data sets and increase the robustness of the model. There are only two kinds of jujube trees in the data set used in this research, so it is necessary to add more kinds of jujube trunk data to enhance the robustness of the model.

(2) Enhance the segmentation ability of the model for small objects. The background of jujube tree is complex, and there are many similar features between the background and jujube tree, which easily leads to false segmentation and missing segmentation. Therefore, the feature fusion ability should be strengthened in the follow-up work to reduce data loss and improve detection accuracy.

(3) In order to further serve intelligent agriculture, this method can be applied to the robot picking operation, providing guidance for the robot to accurately pick fruits.

**Author Contributions:** Data curation, methodology, project administration, writing—original draft, writing—review and editing, Y.Q.; review & editing, Supervision, funding acquisition and project administration, Y.H.; data curation, Z.Z.; formal analysis, Z.Q.; formal analysis, C.W.; data curation, T.G.; review & editing, Supervision, funding acquisition and project administration, J.H. All authors have read and agreed to the published version of the manuscript.

**Funding:** This research was supported by Talent start-up Project of Zhejiang A&F University Scientific Research Development Foundation (2021LFR066), the National Natural Science Foundation of China (C0043619, C0043628).

**Institutional Review Board Statement:** Not applicable.

**Informed Consent Statement:** Not applicable.

**Data Availability Statement:** Not applicable.

**Conflicts of Interest:** The authors declare that they have no conflict of interest to this research.

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
