# Peer review of "A Diameter Measurement Method of Red Jujubes Trunk Based on Improved PSPNet"

_agriculture, doi:10.3390/agriculture12081140_

Round 1
Reviewer 1 Report
The tangent calculations would benefit from a figure on example segmented image.
Camera manufacturer is Intel (not Inter).
The paper appears technically correct and the authors verify their assertion that the proposed approach for jujube trunk segmentation and feature determination has advantages to previous methods. Suggested revisions are to improve the grammar throughout the paper and perform rigorous proofreading. In addition, the text is very dense: it seems that every adjustment and parameter tried has been documented. Two points: is it possible to consolidate these descriptions to make the manuscript more readable? And: the reader would benefit from learning the motivation/rationale behind the parameter adjustments (not just knowing what they were).
Author Response
Response to Reviewers
Dear editors and reviewers:
Thank you very much for reviewing and providing valuable comments and suggestions to our manuscript "A trunk segmentation algorithm of red jujubes based on improved PSPNet and a diameter measurement algorithm". We have carefully revised and marked up the original manuscript in response to your comments and suggestions. For your clarity, revisions to the reviewer 1 were marked in red and those to the reviewer 2 were marked in blue. In the following,we reply and explain point by point on the comments and hope you will be satisfied with our carefully revised manuscript. Please inform us if you think the manuscript needs to modify again. We can revise the manuscript till you and reviewers are satisfied. We look forward to hearing from you.
Kind regards,
Yaohua Hu
Reviewer 1:Comments 1. Camera manufacturer is Intel (not Inter)
Reply: Thank you very much for your question. Inter RealSense D435i has been explained to Intel RealSense D435i (Page 4, line 124, line 133; Page 14, line 384, line 386;)
Reviewer 1:Comments 2. is it possible to consolidate these descriptions to make the manuscript more readable?
Reply: Thank you very much for your question. According to your question, we have improved the grammar and adjusted the content of the manuscript to make it more readable.
Reviewer 1: Comments 3. the reader would benefit from learning the motivation/rationale behind the parameter adjustments (not just knowing what they were).
Reply: Thank you very much for your critical comments. parameters are important to the training convergence of the models, due to our negligence, we have not given the parameters related to the improved algorithm training. Following your suggestion, we have provided the reason of parameter settings. The reasons are as follows:
- Due to thelimitation of training equipment, the batch size was set to 2. If the batch size is too small, the network training procedure is too slow, while if the batchsize is too large it will take up too much memory and make it impossible to train.
- Because the Loss value tends to convergewhen the model was trained to 150 iterations, the iterations was set to 200 in this research. For other parameters, the research referred to reference 33 and the same parameter settings were used in this study (Page 15, Line 415).
References
[33] Ma, B.; Du, J.; Wang, L.; Jiang, H.; Zhou, M. Automatic branch detection of jujube trees based on 3D reconstruction for dormant pruning using the deep learning-based method. Computers and Electronics in Agriculture 2021, 190, 106484.

Reviewer 2 Report
The article presents a software implementation applied to image processing used for measuring a fruit dimension. The title should be changed because it is confusing. From line 116 I understand that a trunk segmentation network is built for mobile terminals. So, it’s more than an algorithm.
The methodology used should be explained better in a clear way.
On line 97 please explain the acronym CNN.
In chapter "Introduction" it’s presented a lot of other works but the chapter is not very clear and doesn’t have coherence. It can be put in a short text/image to summarize the aim of the work. Also, a can be included a classification.
In the chapter "Materials and methods" should be presented a block diagram of the system/implementation for a better understanding.
The figures and tables should have a better description. Also, figures 1, 2, 6, 11, 12, and 15 and Tables 1, 2, 3, 4, and 5 should be modified to respect the template. All the abbreviations in figures 4-9, should be explained (eq. BN, ReLU, etc. )
In eq. 1 and 2 should be explained the functions ??????? and ???????.
The specific parameters in table 1 should be explained
In line 330 the first phrase should be rewritten.
In eq.3 what means x and y?
The “tangent direction” should be replaced with another term because it is confusing. It’s a line, a coordinate, an angle, or has another meaning?
In line 357 “Calculation of the distance between the pixels with diameter” should be rephrased
The chapter "Conclusions" should be rewritten for a better summarizing of the work. How do the authors want to continue their work?
Round 2
Reviewer 2 Report
I think that the article can be published.